# Deterministic Entanglement Swapping with Hybrid Discrete- and Continuous-Variable Systems

**Shou-Bang Yang, Wen Ning, Ri-Hua Zheng, Zhen-Biao Yang \*** and **Shi-Biao Zheng**

Fujian Key Laboratory of Quantum Information and Quantum Optics, College of Physics and Information Engineering, Fuzhou University, Fuzhou 350108, China; ysbscuphy@163.com (S.-B.Y.); n191110012@fzu.edu.cn (W.N.); 201120015@fzu.edu.cn (R.-H.Z.); t96034@fzu.edu.cn (S.-B.Z.)

\* Correspondence: zbyang@fzu.edu.cn

**Abstract:** The study of entanglement between discrete and continuous variables is an important theoretical and experimental topic in quantum information processing, for which entanglement swapping is one of the interesting elements. Entanglement swapping allows two particles without interacting with each other in any way, to form an entangled state by the action of another pair of entangled particles. In this paper, we propose an experimentally feasible scheme to realize deterministic entanglement swapping in the hybrid system with discrete and continuous variables. The process is achieved by preparing two pairs of entangled states, each is formed by a qubit and two quasi-orthogonal coherent state elements of a cavity, performing a Bell-state analysis through nonlocal operations on the continuous variable states of the two cavities, and projecting the two qubits into a maximally entangled state. The present scheme may be applied to other physical systems sustaining such hybrid discrete and continuous forms, providing a typical paradigm for entanglement manipulation through deterministic swapping operations.

**Keywords:** entanglement swapping; discrete variables system; continuous variables system

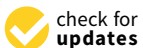

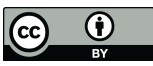

## 1. Introduction

Since the theory of quantum mechanics was proposed, it has never ceased exploring the physical properties of the world. As revealed by Einstein, Podolsky, and Rosen in their seminal paper [1], when two subsystems are prepared in an entangled state, their properties exhibit nonlocal correlations even if they are separated in space. This feature is in sharp contrast with local realism, which assumes that the observable of an object has definite values no matter whether they are measured or not, and they are not affected by events taking place sufficiently far away. The nonlocal correlation can be captured by the violation of Bell's inequalities [2,3]. Besides being central to the fundamentals of quantum mechanics, entanglement is also of great importance in the work of many quantum information tasks such as quantum teleportation [4] and measurement-based quantum computation [5]. Due to such nonlocal properties, two particles initially without any interaction could be put into an entangled state through certain ways of nonlocal manipulation, for which entanglement swapping is one of the unique ways [6]. The sketch of entanglement swapping is shown in Figure 1.

Quantum information processing has been carried out in different types of information encoding, depending on observable variables (the degree of freedom), which can be discrete or continuous [7]. Conditional entanglement swapping has been experimentally demonstrated in photonic qubits [8–17]. In these experiments, swapping of entanglement succeeded probabilistically based on the fact that only a small portion of experimental runs were detected due to photon loss on optical components. Up to 2008, Riebe et al. reported the first experimental demonstration of deterministic, high-fidelity entanglement swapping in ionic qubits [18]. Recently, an experiment has been performed to demonstrate the deterministic entanglement swapping with qubits in a superconducting circuit [19], and further

revealed the striking features of quantum mechanics by using the delayed-choice manner [20]. Experimental demonstrations of unconditional entanglement swapping have been reported in photonic continuous variables [21,22], but the entanglement after the swapping was quite incomplete due to the limitation of entanglement degree of freedom in the initial entangled pairs. Entanglement swapping utilizing hybrid discrete and continuous variables has also been proposed or demonstrated in optical systems [23,24], where the progress has been made to bridge the two approaches that took advantage of their respective intrinsic superiority.

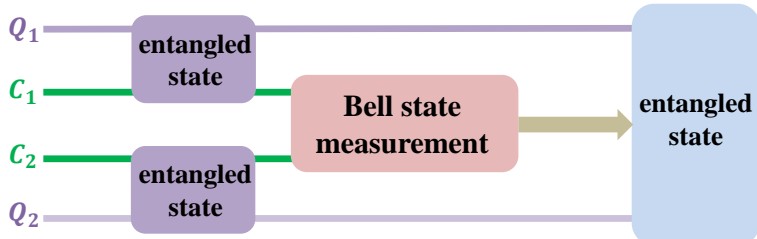

**Figure 1.** Sketch of entanglement swapping. Initially, $Q_1$ is entangled with $C_1$ and $Q_2$ with $C_2$. Bell state measurement on the cavities $C_1$ and $C_2$ will project $Q_1$ and $Q_2$ to one of four orthogonal maximum entangled states.

In this paper, we propose a scheme for the realization of deterministic swapping with hybrid discrete- and continuous- variable systems, embodied, respectively, with qubits and two quasi-orthogonal coherent state elements of two cavity modes. This work is divided into the following steps: (i) Preparation of two pairs of maximally entangled states of hybrid subsystems. (ii) Realization of nonlocal operations on two cavities through the controlled phase gate (CPHASE gate) operation assisted by an ancilla qubit plus two displacement operations on each cavity before and after the CPHASE gate, to convert the entangled type of the quasi-orthogonal coherent state elements involved in the whole system state to the direct product of even and odd cat state elements, before being mapping into two states of the ancilla qubit. (iii) Measurement of the cavity states by first coupling them with the ancilla qubit in succession and then mapping the even (odd) cat state elements of the cavities to the ground (excited) state of the ancilla qubit; the successive twice measurement of the ancilla's states finally projects the two qubits to one of four maximally entangled states. This work can be extended to kinds of physical systems for such or other types of entanglement manipulation [25–31], enriching the ways of quantum state control in hybrid systems.

The paper is organized as follows. In Section 2, we describe the preparation of two pairs of hybrid entangled states of the qubits and the cavity modes. In Section 3, we discuss the Bell state measurement by use of an ancilla qubit, to perform CPHASE gate operation on the two cavities which are entangled with the qubits in the form of the quasi-orthogonal coherent state elements involved in the whole system state, to convert them to the direct product of even or odd cat state elements. In Section 4, we show how these different cat state elements of the cavities can be distinguished by mapping them to the ancilla qubit's states. In Section 5, we give the conclusion.

## 2. Preparation of Hybrid Entangled States in Two Subsystems

In order to get the hybrid qubit–cavity entangled state, we initialize each qubit–cavity subsystem state as

$$|\psi_0\rangle = \frac{1}{\sqrt{2}}(|g\rangle + |e\rangle)|\alpha\rangle. \tag{1}$$

The coupling between the qubit and the cavity in this subsystem is described by the Hamiltonian [32–35]

$$H_0 = -\chi_{QC}|e\rangle\langle e|a^\dagger a, \tag{2}$$

where $\chi_{QC}$ is the effective qubit–cavity coupling strength, which describes the strength of the Stark shifts induced by the dispersive qubit–cavity coupling; $a^\dagger$ and $a$ denoting the raising and lowering operators for the cavity's quantum number.

Under the control of $H_0$ in Equation (2), the state element $|g\rangle|\alpha\rangle$ keeps constant, while $|e\rangle|\alpha\rangle$ evolves as $|e\rangle\left|\alpha e^{i\chi_{QC}t}\right\rangle$. For an interaction time $t = \pi/\chi_{QC}$, the initial state in Equation (1) transforms into the target hybrid entangled state, which we write down, respectively, for $(Q_1, C_1)$ and $(Q_2, C_2)$ as

$$|\psi\rangle_{Q_1 C_1} = \frac{1}{\sqrt{2}}\left(|g\rangle_{Q_1}|\alpha\rangle_{C_1} + |e\rangle_{Q_1}|-\alpha\rangle_{C_1}\right) \tag{3}$$

and

$$|\psi\rangle_{Q_2 C_2} = \frac{1}{\sqrt{2}}\left(|g\rangle_{Q_2}|\alpha\rangle_{C_2} + |e\rangle_{Q_2}|-\alpha\rangle_{C_2}\right). \tag{4}$$

Obviously, in each subsystem the qubit is maximally entangled with the cavity field sustaining the continuous variable coherent state element:

$$|\pm\alpha\rangle = e^{-|\alpha|^2/2} \sum_{n=0}^{\infty} \frac{(\pm\alpha)^n}{\sqrt{n!}}|n\rangle. \tag{5}$$

Thus, the state of the whole system can be written as

$$\begin{aligned}|\psi\rangle &= \frac{1}{2}\big(|\Phi^+\rangle_{Q_1 Q_2}|\Phi^+\rangle_{C_1 C_2} + |\Phi^-\rangle_{Q_1 Q_2}|\Phi^-\rangle_{C_1 C_2}\\ &\quad + |\Psi^+\rangle_{Q_1 Q_2}|\Psi^+\rangle_{C_1 C_2} + |\Psi^-\rangle_{Q_1 Q_2}|\Psi^-\rangle_{C_1 C_2}\big),\end{aligned} \tag{6}$$

with

$$\begin{aligned}|\Phi^\pm\rangle_{Q_1 Q_2} &= \frac{1}{2}\big(|g\rangle_{Q_1}|g\rangle_{Q_2} + |g\rangle_{Q_1}|e\rangle_{Q_2} \pm |e\rangle_{Q_1}|g\rangle_{Q_2} \mp |e\rangle_{Q_1}|e\rangle_{Q_2}\big),\\ |\Psi^\pm\rangle_{Q_1 Q_2} &= \frac{1}{2}\big(|g\rangle_{Q_1}|g\rangle_{Q_2} - |g\rangle_{Q_1}|e\rangle_{Q_2} \pm |e\rangle_{Q_1}|g\rangle_{Q_2} \pm |e\rangle_{Q_1}|e\rangle_{Q_2}\big),\\ |\Phi^\pm\rangle_{C_1 C_2} &= \frac{1}{2}\big(|\alpha\rangle_{C_1}|\alpha\rangle_{C_2} + |\alpha\rangle_{C_1}|-\alpha\rangle_{C_2} \pm |-\alpha\rangle_{C_1}|\alpha\rangle_{C_2} \mp |-\alpha\rangle_{C_1}|-\alpha\rangle_{C_2}\big),\\ |\Psi^\pm\rangle_{C_1 C_2} &= \frac{1}{2}\big(|\alpha\rangle_{C_1}|\alpha\rangle_{C_2} - |\alpha\rangle_{C_1}|-\alpha\rangle_{C_2} \pm |-\alpha\rangle_{C_1}|\alpha\rangle_{C_2} \pm |-\alpha\rangle_{C_1}|-\alpha\rangle_{C_2}\big),\end{aligned} \tag{7}$$

where $\{|\Phi^+\rangle_{Q_1 Q_2}, |\Phi^-\rangle_{Q_1 Q_2}, |\Psi^+\rangle_{Q_1 Q_2}, |\Psi^-\rangle_{Q_1 Q_2}\}$ and $\{|\Phi^+\rangle_{C_1 C_2}, |\Phi^-\rangle_{C_1 C_2}, |\Psi^+\rangle_{C_1 C_2}, |\Psi^-_{34}\rangle_{C_1 C_2}\}$ represent two groups of orthogonally entangled state subspaces formed by the qubits and the cavity modes, respectively.

## 3. The State Conversion of the Two Cavities: From the Entangled State Elements to the Direct Product Elements

To realize entanglement swapping in this hybrid system, we need to perform a Bell-state measurement on the two cavities, achieved by coupling dispersively to them with a driven ancilla qubit ($Q_a$), to induce a geometric $\pi$-phase [36]. The interaction Hamiltonian for describing the dispersive coupling between the two cavities and the ancilla qubit can be modeled by [37,38]

$$H = -\chi_{Q_a C_1}|e\rangle_{Q_a}\langle e|a^\dagger_{C_1} a_{C_1} - \chi_{Q_a C_2}|e\rangle_{Q_a}\langle e|a^\dagger_{C_2} a_{C_2} \tag{8}$$

where $\chi_{Q_a C_1}$ ($\chi_{Q_a C_2}$) is the dispersive coupling strength between $Q_a$ and $C_1$ ($C_2$).

To realize the geometric CPHASE gate for the two cavity states, we use a microwave field to address the ancilla qubit. The interaction Hamiltonian for the ancilla qubit resonantly coupling to the driven microwave field is written as

$$H_d = \frac{1}{2}\varepsilon(e^{i\phi}|e\rangle_{Q_a}\langle g| + e^{-i\phi}|g\rangle_{Q_a}\langle e|), \tag{9}$$

where $\varepsilon$ and $\phi$ are the Rabi frequency and phase of the driven field, respectively. The validity of the Equation (9) is conditional on the vacuum states for the two cavities. As the ancilla qubit is simultaneously coupling to the two cavities, the validity of the geometric CPHASE gate operation should be guaranteed by the condition that the Rabi frequency $\varepsilon$ is much smaller than $\bar{n}_{C_j}\chi_{Q_aC_j}$, where $n_{C_j}$ $(j = 1, 2)$ is the average photon number for the cavity $C_j$. When there are photons in any cavity, the dispersive coupling to the ancilla qubit produces stark shifts to the naked frequency of resonance, disabling the geometric CPHASE gate operation. The effective Hamiltonian considering such a conditional constraint can be written as

$$\begin{aligned} H_{eff} &= \frac{1}{2}\varepsilon e^{i\phi}e^{-i\chi_{Q_aC_1}a^{\dagger}_{C_1}a_{C_1}}e^{-i\chi_{Q_aC_2}a^{\dagger}_{C_2}a_{C_2}}|e\rangle_{Q_a}\langle g| + H.c. \\ &\approx \frac{1}{2}\varepsilon e^{i\phi}|e\rangle_{Q_a}\langle g| \otimes |0\rangle_{C_1}\langle 0| \otimes |0\rangle_{C_2}\langle 0| + H.c.. \end{aligned} \tag{10}$$

Under the action of the Hamiltonian in Equation (10) for an operation time $t = 2\pi/\varepsilon$, the joint qubit–cavity state element $|e\rangle_{Q_a}|0\rangle_{C_1}|0\rangle_{C_2}$ transforms to $-|e\rangle_{Q_a}|0\rangle_{C_1}|0\rangle_{C_2}$, while with all other elements remaining unchanged, which is key to the CPHASE gate operation for the two cavities.

In order to use such a CPHASE gate for the Bell-state measurement on the two cavities, we need to sandwich such a gate operation with two mutually inverse displacement operations $D_{C_j}(\alpha)$ and $D^{\dagger}_{C_j}(\alpha)$ $(j = 1, 2)$ to each cavity. The combination of such displacement operations and CPHASE gate operations can be unified as

$$P^{\pi}_{C_j} = D_{C_j}(\alpha)R^{\pi}D^{\dagger}_{C_j}(\alpha), \tag{11}$$

with

$$D_{C_j}(\alpha) = e^{\alpha a^{\dagger} - \alpha^* a} \tag{12}$$

being the cavity displacement operator.

After the transformation of $P^{\pi}_{C_j}$, the entangled state elements $|\Phi^{\pm}\rangle_{C_1C_2}$ and $|\Psi^{\pm}\rangle_{C_1C_2}$ in Equation (7), respectively, change to

$$\begin{aligned} |\Phi'^{\pm}\rangle_{C_1C_2} &= \frac{1}{2}(|\alpha\rangle_{C_1}|\alpha\rangle_{C_2} + |\alpha\rangle_{C_1}|-\alpha\rangle_{C_2} \pm |-\alpha\rangle_{C_1}|\alpha\rangle_{C_2} \pm |-\alpha\rangle_{C_1}|-\alpha\rangle_{C_2}) \\ &\equiv \frac{1}{2}(|\alpha\rangle \pm |-\alpha\rangle)_{C_1}(|\alpha\rangle + |-\alpha\rangle)_{C_2}, \\ |\Psi'^{\pm}\rangle_{C_1C_2} &= \frac{1}{2}(|\alpha\rangle_{C_1}|\alpha\rangle_{C_2} - |\alpha\rangle_{C_1}|-\alpha\rangle_{C_2} \pm |-\alpha\rangle_{C_1}|\alpha\rangle_{C_2} \mp |-\alpha\rangle_{C_1}|-\alpha\rangle_{C_2}) \\ &\equiv \frac{1}{2}(|\alpha\rangle \pm |-\alpha\rangle)_{C_1}(|\alpha\rangle - |-\alpha\rangle)_{C_2}. \end{aligned} \tag{13}$$

Thus, the hybrid entangled states $|\psi_t\rangle$ can now be expanded as

$$\begin{aligned} |\psi_t\rangle &= \frac{1}{2}(|\Phi^+\rangle_{Q_1Q_2} \otimes |\mathcal{C}_+\rangle_{C_1}|\mathcal{C}_+\rangle_{C_2} + |\Psi^+\rangle_{Q_1Q_2} \otimes |\mathcal{C}_+\rangle_{C_1}|\mathcal{C}_-\rangle_{C_2} \\ &\quad + |\Phi^-\rangle_{Q_1Q_2} \otimes |\mathcal{C}_-\rangle_{C_1}|\mathcal{C}_+\rangle_{C_2} + |\Psi^-\rangle_{Q_1Q_2} \otimes |\mathcal{C}_-\rangle_{C_1}|\mathcal{C}_-\rangle_{C_2}), \end{aligned} \tag{14}$$

where

$$|\mathcal{C}_+\rangle = \mathcal{N}_e(|\alpha\rangle + |-\alpha\rangle),$$
$$|\mathcal{C}_-\rangle = \mathcal{N}_o(|\alpha\rangle - |-\alpha\rangle), \tag{15}$$

represent the even and odd Schrödinger cat states, respectively, and

$$\mathcal{N}_{e,o} = \frac{1}{\sqrt{2}}(1 \pm e^{-2|\alpha|^2})^{-1/2} \tag{16}$$

are the normalization coefficients.

Obviously, the four Bell-state elements of the two qubits are, respectively, correlated with four combinations of direct product of even or odd cat state elements of the two cavities.

## 4. Distinguishing the States of the Two Cavities

The next step is to distinguish the four combinations of the cavity state elements as to collapse the two qubits to a specific Bell state.

We begin by coupling the cavity $C_1$ with the ancilla qubit $Q_a$ and write down the state elements for $C_1$ and $Q_a$ involved in the whole system state as

$$
\begin{aligned}
|\psi'_+\rangle &= \frac{1}{\sqrt{2}}|g\rangle_{Q_a}|\mathcal{C}_+\rangle_{C_1}, \\
|\psi'_-\rangle &= \frac{1}{\sqrt{2}}|g\rangle_{Q_a}|\mathcal{C}_-\rangle_{C_1},
\end{aligned} \tag{17}
$$

where the parity combination of the two cavity state elements determines the four different entangled states of the qubits. We then apply a Hadamard gate on the qubit $Q_a$ to turn these elements to

$$
\begin{aligned}
|\psi''_+\rangle &= \frac{1}{\sqrt{2}}(|g\rangle + |e\rangle)_{Q_a}|\mathcal{C}_+\rangle_{C_1}, \\
|\psi''_-\rangle &= \frac{1}{\sqrt{2}}(|g\rangle + |e\rangle)_{Q_a}|\mathcal{C}_-\rangle_{C_1}.
\end{aligned} \tag{18}
$$

Under the evolution of the Hamiltonian of Equation (8) with $t = \pi/\chi_{Q_a C_1}$, these elements in (18) further change to

$$
\begin{aligned}
|\psi''_+\rangle &= \frac{1}{\sqrt{2}}(|g\rangle + |e\rangle)_{Q_a}|\mathcal{C}_+\rangle_{C_1}, \\
|\psi''_-\rangle &= \frac{1}{\sqrt{2}}(|g\rangle - |e\rangle)_{Q_a}|\mathcal{C}_-\rangle_{C_1}.
\end{aligned} \tag{19}
$$

We then apply a second Hadamard gate to the qubit, which projects the even/odd cat element of the two cavities to the qubit's $|g\rangle$ and $|e\rangle$ state. Thus, the measurement of the qubit's state distinguishes the state of the cavity $C_1$. Note here that when we use the dispersive interaction between the ancilla qubit and the cavity $C_1$ described by the Hamiltonian in Equation (8), we should disable the dispersive interaction between the ancilla qubit and the cavity $C_2$, which can be solved by minor improvement of the recently demonstrated circuit quantum electrodynamics techniques [36–38]. The same case is applied to the second time for this cavity parity information analysis ($C_1 \leftrightarrow C_2$).

Recycling the qubit and repeating the same operations for the second time, we can further distinguish the parity information of the state of the cavity $C_2$. The whole mapping process is given as follows:

$$
\begin{aligned}
|g\rangle_{1st}|g\rangle_{2nd} &\to |\mathcal{C}_+\rangle_{C_1}|\mathcal{C}_+\rangle_{C_2}, \\
|g\rangle_{1st}|e\rangle_{2nd} &\to |\mathcal{C}_+\rangle_{C_1}|\mathcal{C}_-\rangle_{C_2}, \\
|e\rangle_{1st}|g\rangle_{2nd} &\to |\mathcal{C}_-\rangle_{C_1}|\mathcal{C}_+\rangle_{C_2}, \\
|e\rangle_{1st}|e\rangle_{2nd} &\to |\mathcal{C}_-\rangle_{C_1}|\mathcal{C}_-\rangle_{C_2}.
\end{aligned}
\tag{20}
$$

Thus, the twice qubit's state measurements get exactly the parity combination of the two cavities and collapse the two qubits to a specific entangled state as described in Equation (7). For the purpose of intuition, we further apply a Hadamard gate on $Q_2$: $(|g\rangle \to (|g\rangle + |e\rangle)/\sqrt{2}; |e\rangle \to (|g\rangle - |e\rangle)/\sqrt{2})$, and change these qubits' entangled states to the following standard Bell states:

$$
\begin{aligned}
|\Phi^+\rangle_{Q_1 Q_2} &= \frac{1}{\sqrt{2}}(|g\rangle_{Q_1}|g\rangle_{Q_2} + |e\rangle_{Q_1}|e\rangle_{Q_2}), \\
|\Psi^+\rangle_{Q_1 Q_2} &= \frac{1}{\sqrt{2}}(|g\rangle_{Q_1}|e\rangle_{Q_2} + |e\rangle_{Q_1}|g\rangle_{Q_2}), \\
|\Phi^-\rangle_{Q_1 Q_2} &= \frac{1}{\sqrt{2}}(|g\rangle_{Q_1}|g\rangle_{Q_2} - |e\rangle_{Q_1}|e\rangle_{Q_2}), \\
|\Psi^-\rangle_{Q_1 Q_2} &= \frac{1}{\sqrt{2}}(|g\rangle_{Q_1}|e\rangle_{Q_2} - |e\rangle_{Q_1}|g\rangle_{Q_2}).
\end{aligned}
\tag{21}
$$

Note that this local single-qubit operation does not alter the entanglement degree of the two qubits' state. The operation sequences are figured out and shown in Figure 2.

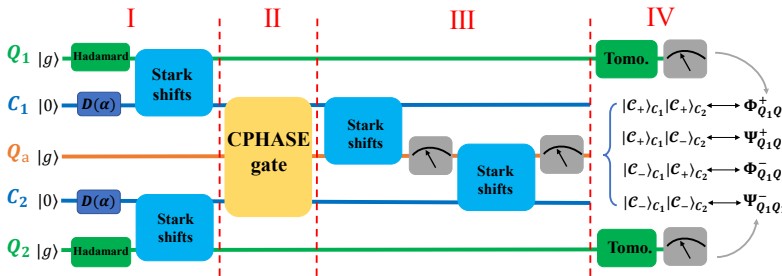

**Figure 2.** Simulated experimental sequences. The procedure consists of four parts: (**I**) Generation of entangled states for $Q_1, C_1$ and $Q_2, C_2$ through the Stark shifts induced by dispersive qubit–cavity interaction; (**II**) application of a CPHASE gate operation on the two cavities to transform the entangled coherent state elements into the direct product of even or odd cat elements; (**III**) distinguishing the four combinations of cat states of the two cavities to collapse the two qubits into a specific entangled state; and (**IV**) reconstruction of the density matrices for $Q_1$ and $Q_2$, which can be achieved by standard process of quantum state tomography, as reported in the previously demonstrated experiments [19,39,40].

After the analysis of the states of cavities, we extract the density matrix of joint 2-qubit state for $Q_1$ and $Q_2$, which is achieved through the standard process of quantum state tomography [19,39,40]. The measured density matrices of $Q_1$ and $Q_2$, conditional on the measurement outcomes $|g\rangle_{1st}|g\rangle_{2nd}$, $|g\rangle_{1st}|e\rangle_{2nd}$, $|e\rangle_{1st}|g\rangle_{2nd}$, and $|e\rangle_{1st}|e\rangle_{2nd}$ of the twice measurement of $Q_a$ corresponding to $|\mathcal{C}_+\rangle|\mathcal{C}_+\rangle$, $|\mathcal{C}_+\rangle|\mathcal{C}_-\rangle$, $|\mathcal{C}_-\rangle|\mathcal{C}_+\rangle$, and $|\mathcal{C}_-\rangle|\mathcal{C}_-\rangle$ of $C_1$ and $C_2$, are displayed in Figure 3, where the black wire frames denote the matrix elements of the ideal states.

We perform numerical simulations to check the availability of the proposed entanglement swapping scheme utilizing the Markovian master equations [41]. Our numerical calculations aim to quantitatively characterize the validity of the proposed scheme by

considering the dissipation factors (qubits' energy relaxation and dephasing, and cavity photon's loss and dephasing) and the CPHASE gate operation's errors, but ignoring the possible imperfections in the single-qubit operations and the pulses' time. We adopt the recently demonstrated experimental parameters as reported in [36], which are shown in Table 1. We find that the fidelities for the four Bell states of the qubits are $F_{\Phi_+} = 0.8509$, $F_{\Psi_+} = 0.8848$, $F_{\Phi_-} = 0.8917$, and $F_{\Psi_-} = 0.7891$, respectively, with the corresponding concurrences of $C_{\Phi_+} = 0.7323$, $C_{\Psi_+} = 0.7808$, $C_{\Phi_-} = 0.7937$, and $C_{\Psi_-} = 0.6735$, respectively. The success probabilities for getting these Bell state are $P(\Phi_+) = 0.2554$, $P(\Psi_+) = 0.2481$, $P(\Phi_-) = 0.2463$, and $P(\Psi_-) = 0.2502$, respectively. The errors accumulate during the process of the CPHASE gate operation (the fidelity of which is individually calculated to be 0.9261) mainly result from the qubits' energy relaxation and dephasing and the non-ideal CPHASE gate condition ($\varepsilon << \bar{n}_{C_j}\chi_{Q_aC_j}$). All the errors thus result in the deviation of the two qubits' density matrix elements from the ideal case through the entanglement swapping, as shown in Figure 3. For the improved qubits' parameters: $T_1 = T_\varphi/2 = 50$ µs, the fidelities of these Bell states will be $F_{\Phi_+} = 0.8801$, $F_{\Psi_+} = 0.8843$, $F_{\Phi_-} = 0.9438$, and $F_{\Psi_-} = 0.8695$, respectively. For a perfect CPHASE gate operation and the experimental parameters in Table 1, the fidelities of the entangled states will improve to 0.9281, 0.8913, 0.9356, and 0.8064, respectively. It must be mentioned that inaccuracies in both the single-qubit control and the pulses' time will also reduce the effect of entanglement swapping. For instance, when dealing with the CPHASE gate operation, if there exists a fluctuation of −10% (+10%) of the whole duration time during the gate operation process, it will lead to 2% (3%) reduction in the gate fidelity. Notice that according to the recently demonstrated techniques in the superconducting circuit [36,38], the influences caused by these two aspects could be limited to the minimum range.

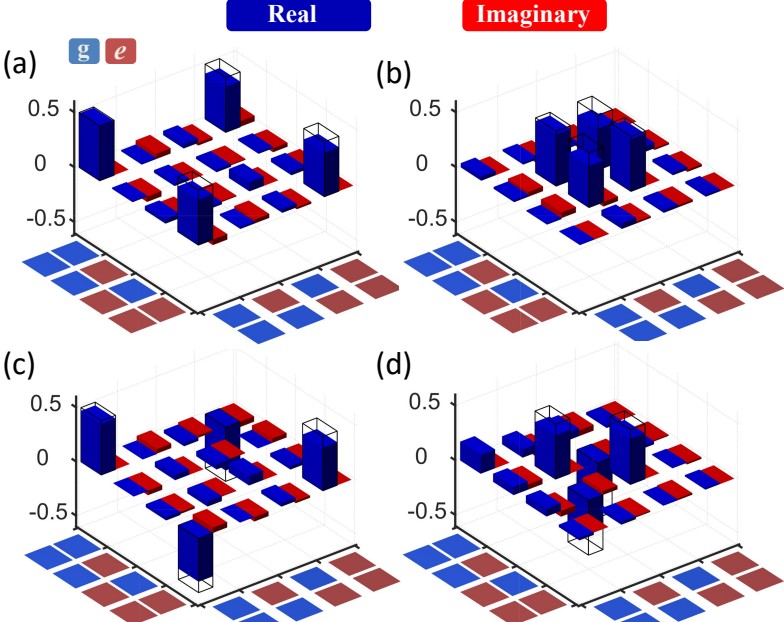

**Figure 3.** Simulated tomography of $Q_1 - Q_2$ density matrices conditional on the four double-measurement outcomes of $Q_a$, which distinguish the states of the two cavities in order to project ($Q_1$, $Q_2$) to a specific entangled state. (**a**) $|g\rangle_{1st}|g\rangle_{2nd} \rightarrow |\mathcal{C}_+\rangle_{C_1}|\mathcal{C}_+\rangle_{C_2} \rightarrow |\Phi^+\rangle_{Q_1Q_2}$; (**b**) $|g\rangle_{1st}|e\rangle_{2nd} \rightarrow |\mathcal{C}_+\rangle_{C_1}|\mathcal{C}_-\rangle_{C_2} \rightarrow |\Psi^+\rangle_{Q_1Q_2}$; (**c**) $|e\rangle_{1st}|g\rangle_{2nd} \rightarrow |\mathcal{C}_-\rangle_{C_1}|\mathcal{C}_+\rangle_{C_2} \rightarrow |\Phi^-\rangle_{Q_1Q_2}$; (**d**) $|e\rangle_{1st}|e\rangle_{2nd} \rightarrow |\mathcal{C}_-\rangle_{C_1}|\mathcal{C}_-\rangle_{C_2} \rightarrow |\Psi^-\rangle_{Q_1Q_2}$.

**Table 1. The characteristics of the qubits and cavities.** $T_1$ and $T_\varphi$ are the energy relaxation time and the dephasing time, and $\chi_{C_1 Q_i}$ ($\chi_{C_2 Q_i}$) are the dispersive coupling strength between cavity $C_1$ ($C_2$) and qubit $Q_j$. Data are taken from [36].

| | Qubit 1 | Qubit 2 | Qubit a | Cavity 1 | Cavity 2 |
|---|---|---|---|---|---|
| $T_1$ (μs) | 35 | 20 | 25 | 480 | 692 |
| $T_\varphi$ (μs) | 39 | 17 | 50 | 1338 | 402 |
| $\chi_{C_1 Q_j}/2\pi$ (MHz) | 1.599 | | 0.524 | | |
| $\chi_{C_2 Q_j}/2\pi$ (MHz) | | 2.670 | 1.494 | | |

## 5. Conclusions

In summary, we have proposed a scheme for the realization of deterministic entanglement swapping for hybrid discrete- and continuous- variable systems. The process mainly includes three steps: (1) Preparing two pairs of qubit–cavity entangled states, each is formed by a qubit and two quasi-orthogonal coherent state elements of a cavity; (2) performing a Bell-state measurement by the CPHASE gate operation on the continuous variable elements of the two cavities; (3) mapping the cavities' parity information to an ancilla qubit for measurements, which projects the two qubits into one of the four maximally entangled state. Our numerical simulations show that entanglement swapping with considerable fidelity can in principle be accomplished with the recently available experimental parameters. The scheme may be extended to other physical systems for entanglement swapping or other types of entanglement manipulation, using such hybrid discrete and continuous variables [7,23,24,42,43].

**Author Contributions:** All contributed to the development of the conceptualization, discussed the results, and commented on the manuscript. All authors have read and agreed to the published version of the manuscript.

**Funding:** This research was funded by the National Natural Science Foundation of China under grant number 11874114 and number 11875108.

**Institutional Review Board Statement:** Not applicable.

**Informed Consent Statement:** Not applicable.

**Data Availability Statement:** Not applicable.

**Conflicts of Interest:** The authors declare no conflict of interest.

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
