# Peer review of "Deterministic Entanglement Swapping with Hybrid Discrete- and Continuous-Variable Systems"

_photonics, doi:10.3390/photonics9060368_

Round 1
Reviewer 2 Report
The manuscript "Deterministic entanglement swapping with hybrid discrete- and continuous-variable systems," which was presented to me for a review, is a valuable contribution to theoretical photonics. The authors introduce a high-fidelity framework for entanglement swapping. The work is divided into three main sections. In Sec. 2, the authors describe the preparation of hybrid entangled states that consist of a qubit and cavity. Then, in Sec. 3, the authors provide operators that allow performing the CPHASE gate operation on the two cavities. Next, in Sec. 4, the authors present a scheme to differentiate between the cavity states. This procedure leads to the collapse of the compound state Eq. (15) into a specific two-qubit entangled state from Eq. (8). Finally, by applying a Hadamard gate, we can get an entangled state that is a type of Bell state. The key message of the manuscript is that through the series of operations presented in the work, we switch from the qubit-cavity entanglement as in Eq. (5-6) to two-qubit entanglement as in Eq. (22).
The manuscript presents an interesting model of entanglement swapping, which deserves publication. However, some minor issues should be addressed before the final decision.
A. I would recommend changing the order of content in Sec. 2. Now, it is a little bit confusing because first, you give the entangled state and the actual initial state appears later. I propose to explain step by step this procedure. So please, start with the actual state that we can prepare out of hand. Then, explain all the steps necessary to get the desired entangled state. Please, pay more attention to how entanglement is induced from an initially separable state Eq. (3). This is crucial and should be elaborated on.
B. At a few points, you consider very specific periods of time. For example, in lines: 73, 99, and 119, we have concrete values of t. In time-continuous processes, this allows you to obtain specific output states. However, there is a hidden assumption that we can precisely control time. If we talk about the feasibility of this protocol, it would be problematic to tune into the required evolution time. Thus, I would appreciate it if you could expand the Conclusions to discuss the limiting factors of the scheme.
C. Also, I have some doubts concerning the state estimation you performed to evaluate the framework. We have the results in the last paragraph in Sec. 4, but I do not find a clear description of your methodology. Please, elaborate on this issue and provide more details about the method of state estimation that you used. Secondly, I am concerned about the way you provide specific results. I would expect to have some uncertainties related to the figures of merit, such as the concurrence. Can you estimate an error margin for each figure? Finally, in line 148, you say "qualitatively". Perhaps, you meant "quantitatively".
In summary, the manuscript is promising. It can be accepted for publication
after the minor issues are resolved.
